# FutureSightDrive: Thinking Visually with Spatio-Temporal CoT for Autonomous Driving

**Shuang Zeng**[1,2][*], **Xinyuan Chang**[2], **Mengwei Xie**[2], **Xinran Liu**[2],
**Yifan Bai**[1,3], **Zheng Pan**[2], **Mu Xu**[2], **Xing Wei**[1][†]

[1]Xi'an Jiaotong University  [2]Amap, Alibaba Group  [3]DAMO Academy, Alibaba Group

zengshuang@stu.xjtu.edu.cn, weixing@mail.xjtu.edu.cn,
{changxinyuan.cxy, xiemengwei.xmw, tom.lxr}@alibaba-inc.com,
{baiyifan.byf, panzheng.pan, xumu.xm}@alibaba-inc.com

## Abstract

Vision–Language–Action (VLA) models are increasingly used for end-to-end driving due to their world knowledge and reasoning ability. Most prior work, however, inserts *textual* chains-of-thought (CoT) as intermediate steps tailored to the current scene. Such symbolic compressions can blur spatio-temporal relations and discard fine visual cues, creating a cross-modal gap between perception and planning. We propose **FSDrive**, a *visual* spatio-temporal CoT framework that enables VLAs to think in images. The model first acts as a world model to generate a **unified future frame** that overlays coarse but physically-plausible priors—future lane dividers and 3D boxes—on the predicted future image. This unified frame serves as the visual CoT, capturing both spatial structure and temporal evolution. The same VLA then functions as an inverse-dynamics model, planning trajectories from current observations and the visual CoT. To equip VLAs with image generation while preserving understanding, we introduce a **unified pre-training paradigm** that expands the vocabulary to include visual tokens and jointly optimizes VQA (for semantics) and future-frame prediction (for dynamics). A progressive easy-to-hard scheme first predicts lane/box priors to enforce physical constraints, then completes full future frames for fine details. On nuScenes and NAVSIM, FSDrive improves trajectory accuracy and reduces collisions under both ST-P3 and UniAD metrics, and attains competitive FID for future-frame generation despite using lightweight autoregression. It also advances scene understanding on DriveLM. Together, these results indicate that **visual CoT** narrows the cross-modal gap and yields safer, more anticipatory planning. Code is available at https://github.com/MIV-XJTU/FSDrive.

## 1  Introduction

Recently, given the superior capabilities of multimodal large language models (MLLMs) in world knowledge, reasoning ability, and interpretability, they have been widely applied in autonomous driving [20, 43, 87, 31]. One promising direction is the end-to-end vision-language-action (VLA) model, which leverages pre-trained vision-language model (VLM) to directly extract scene features from visual observations and language instructions, subsequently generating vehicle control commands (e.g., speed and trajectory). This paradigm not only simplifies system architecture and minimizes information loss, but also enables the utilization of the model's world knowledge to analyze driving environments and reason about safe decisions in complex scenarios.

---

[*] Work done during the internship at Amap, Alibaba Group.

[†] Corresponding author.

39th Conference on Neural Information Processing Systems (NeurIPS 2025).

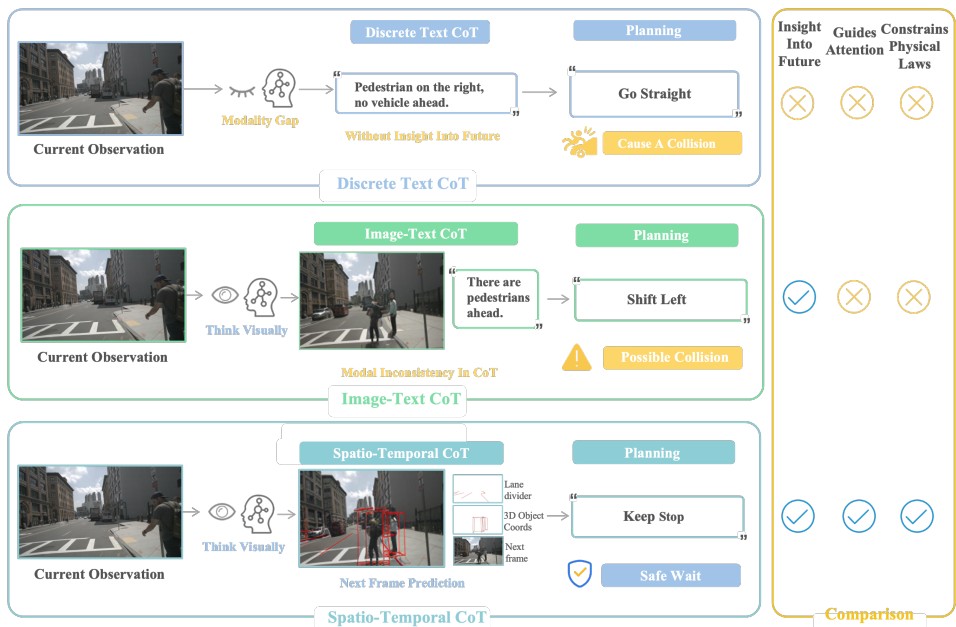

Figure 1: Comparison of different CoT. Textual CoT expression provides insufficient information. The modalities between the image-text CoT are inconsistent. The proposed spatio-temporal CoT captures the temporal and spatial relationships in the future.

In the field of language, Chain-of-Thought (CoT) [67, 50, 15, 52] improves reasoning capabilities and interpretability by encouraging step-by-step thinking. However, existing autonomous driving [27, 44, 14] typically incorporate discrete text CoT (e.g., language descriptions targeting current scenarios and bounding box coordinates) as intermediate reasoning steps. This method is essentially highly abstract and symbolized compression of visual information, which may lead to ambiguous temporal and spatial relationships, loss of fine-grained information, and modality conversion gaps [46, 55, 72], as shown in the top of Figure 1. For autonomous vehicles requiring deep physical-world interaction, should their thinking process more closely resemble simulation and imagination of world, rather than merely relying on logical deduction of language?

Inspired by the human driver's cognitive mechanism of directly constructing visual representations of future scenarios in the mind, rather than converting them into language descriptions for reasoning, we propose a more intuitive spatio-temporal CoT method as shown in the bottom part of Figure 1. This method avoids information loss during text abstraction and enables the model to think visually about trajectory planning. First, the VLA serves as a world model to generate unified image frame for predicting future world states: Inspired by visual prompting engineering [53, 81] that draws red circles on images to guide model attention and by VLIPP [78] first predicts future bounding boxes to introduce physical priors when generating future frames, we represent future world spatial relationships through future red lane dividers and 3D detection boxes on the predicted unified frames [80]. These coarse-grained visual cues direct the model's attention toward drivable areas and critical objects in future scenes while enforcing physically plausible constraints. Meanwhile, the temporal relationships are represented by the ordinary future frame, where the dynamic evolution of visual content intuitively characterizes temporal progression and the inherent laws of scene development. Subsequently, the spatio-temporal CoT acts as an intermediate reasoning step, enabling the VLA to function as an inverse dynamics model for trajectory planning based on current observations and future predictions. Compared to traditional discrete text CoT, and even image-text CoT methods [27, 91, 41] as shown in the middle of the Figure 1, our method unifies both future scene representations and perception outputs in image format, which more effectively conveys the temporal and spatial relationships. This eliminates semantic gaps caused by cross-modal conversions (e.g., converting visual perceptions into textual descriptions for reasoning), establishing an end-to-end visual reasoning pipeline that enables direct visual causal inference by the model.

To endow VLAs with image generation capabilities, we propose a pre-training paradigm that simultaneously preserves the semantic understanding of existing MLLM and activates their visual

generation capacity. Specifically, for the semantic understanding preservation part, we follow previous approaches [64, 27, 25] by incorporating visual question answering (VQA) tasks for current scene comprehension. For the activation of visual generation capabilities, we investigate the shared vocabulary space between image and text, directly unleashing the visual generation potential of existing MLLMs in the field of autonomous driving through minimal data (approximately 0.3% of previous methods [70, 73, 24, 35]) without requiring complex model architecture modifications or redesigns. However, directly generating complete detailed future scenes may fail to adhere to physical laws [78, 88]. Thus, we propose a progressive, easy-to-hard generation method. We leverage the world knowledge of VLAs to first infer drivable regions and key object positions in future scenarios, generating coarse-grained future perception images (e.g., lane dividers and 3D detection) to constrain physical laws. Subsequently, full future frames are generated under this constraint to supplement fine-grained details, enabling the model to think visually about accurate future prediction.

Extensive experiments on trajectory planning, future frames generation, and scene understanding tasks demonstrate the effectiveness of pre-training paradigm and spatio-temporal CoT in FSDrive. FSDrive achieves road scene comprehension by establishing pixel-level embodied associations with the environment, rather than relying on human-designed abstract linguistic symbols, advancing autonomous driving towards visual reasoning. In summary, our main contributions are as follows:

- We propose a spatio-temporal CoT reasoning method that allows the model to enhance trajectory planning by thinking visually from future temporal and spatial dimensions.

- We propose a unified pre-training paradigm for visual generation and understanding. Meanwhile, we introduce a progressive generation approach that evolves from imposing physical constraints to supplementing details.

- We conduct comprehensive evaluations across trajectory planning, future frames generation, and scene understanding tasks, demonstrating the effectiveness of our FSDrive.

## 2 Related work

### 2.1 Unified multimodal understanding and generation

Recent research efforts [38, 70, 49, 68] have increasingly focused on unifying multimodal understanding and visual generation within a single LLM. On one front, methods like Show-o [74], and VILA-U [73] employ VQ-VAE [61] to transform images into discrete tokens while training LLMs to predict them. However, these methods suffer from insufficient semantic information preservation, often leading to performance degradation in downstream understanding tasks. Alternative methods [57, 11, 48, 9, 82] utilize ViT [12]-based vision encoders (e.g., CLIP [51]) to encode images into continuous feature maps. Nevertheless, such methods typically depend on external diffusion models for image generation or use different training objectives (i.e. diffusion and autoregression) for the two tasks, further complicates the infrastructure design with overall lower efficiency. Moreover, the aforementioned methods usually require massive billion-scale datasets for extensive training from scratch, which results in prohibitively high computational costs when disseminating explorations in this form. In this work, we demonstrate that the visual generative capabilities of existing MLLMs can be directly activated through minimal training costs (approximately 0.3% of previous methods [70, 58, 42, 8]) without requiring sophisticated architectural designs.

### 2.2 Vision-language models for autonomous driving

Given the superior capabilities of large language models (LLMs) in world knowledge, reasoning, and interpretability, recent researches [2, 83, 39, 85] increasingly integrate Vision-Language Models (VLMs)/LLMs with autonomous driving systems to address limitations in end-to-end approaches. DriveGPT4 [76] employs LLMs through iterative question-answering interactions to explain vehicle behaviors and predict control signals. DriveVLM [60] synergizes LLMs with end-to-end architectures, where LLMs predict low-frequency trajectories that are subsequently refined by the end-to-end model for final planning. Doe-1 [95] reformulates autonomous driving as a next-token prediction task using Lumina-mGPT's [37] multimodal generation capabilities, executing diverse tasks through multimodal token processing. EMMA [27] leverages Gemini's multimodal foundation by encoding all non-sensor inputs (navigation instructions, vehicle status) and outputs (trajectories, 3D positions) as natural language text, fully exploiting pre-trained LLMs' world knowledge. In this work, we propose a

spatio-temporal chain of thought (CoT) reasoning method that unifies the form of images, allowing the model to think visually about trajectory planning.

## 2.3 World models for autonomous driving

World models [66, 45, 90, 89] aim to infer ego status and dynamic environments from past observations to enable accurate future prediction and planning. Current applications of world models in autonomous driving primarily focus on driving scenario generation [47, 16, 32], planning [66, 41], and representation learning [45, 79, 84]. For driving scenario generation, most prior works are built upon diffusion models, with the exception of GAIA-1 [18] which incorporates a progressive next-token predictor and an additional diffusion image decoder. Recent DrivingGPT [5] leverages existing vision generation LLM LlamaGen [56] while simultaneously outputting predictions for future states and actions. However, such VQ-VAE based visual tokens lack semantic information, often leading to performance degradation in downstream visual understanding tasks [74, 40, 59]. In this work, we propose to directly activate the visual generation capabilities of existing multimodal large language models, enabling VLMs to act as world models and predict future frames.

## 3 Proposed method: FSDrive

The proposed FSDrive is illustrated in Figure 2. Section 3.1 describes the preliminaries. Section 3.2 presents a unified visual generation and understanding pre-training paradigm and a progressive generation method. Section 3.3 proposes spatio-temporal chain-of-thought methods. Section 3.4 details the training strategy.

### 3.1 Preliminary

**End-to-end trajectory planning.** End-to-end autonomous driving directly generates future trajectory from sensor data, convertible to vehicle control actions like acceleration and steering [27]. given $N$ surround-view images $I_t = \{I_t^1, I_t^2, \ldots, I_t^N\}$ at timestep $t$, model $\mathcal{M}$ outputs a BEV trajectory $W_t = \{w_t^1, w_t^2, \ldots, w_t^n\}$, where each waypoint $w_t^i = (x_t^i, y_t^i)$. The process is formulated as:

$$W_t = \mathcal{M}(I_t, opt(T_{com}, T_{ego})), \tag{1}$$

$opt(T_{com}, T_{ego})$ denotes optional navigation commands and ego status (e.g., velocity, acceleration).

**Unified visual generation and understanding.** Recent works [70, 22] unify multimodal understanding and vision generation in single LLM. While understanding aligns with standard LLMs, generation methods [38, 23] typically use VQ-VAE [61] to encode images into discrete tokens. First, the image tokenizer quantizes image pixels $x \in \mathbb{R}^{H \times W \times 3}$ into discrete tokens $q \in \mathcal{Q}^{h \times w}$, where $h = H/p$, $w = W/p$, $p$ is the downsampling factor, and $q(i, j)$ represents the index of the image codebook. These $h \cdot w$ tokens are arranged in raster order to train a Transformer [62]-based autoregressive model. During image generation, a general language modeling (LM) objective is adopted to autoregressively predict the next token, maximizing the likelihood of each image token:

$$\mathcal{L} = -\sum_{i=1} \log P_\theta(q_i|q_{<i}), \tag{2}$$

where $q_i$ denotes the visual token and $\theta$ represents the LLM parameters. Finally, the VQ-VAE's detokenizer converts these image tokens back into image pixels.

### 3.2 Unified pre-training paradigm for visual generation and understanding

To enable unified pre-training, MLLMs require visual generation capabilities. As described in Section 3.1, existing methods (e.g. Lumina-mGPT [37], the visual generation LLM used by Doe-1 [95]) typically employ VQ-VAE to encode images into discrete tokens when extracting visual information. However, these tokens lack semantic information, which hurts downstream understanding performance [74, 97]. Moreover, current methods [70, 96] demand expensive training from scratch on massive billion-scale datasets without leveraging existing LLM knowledge.

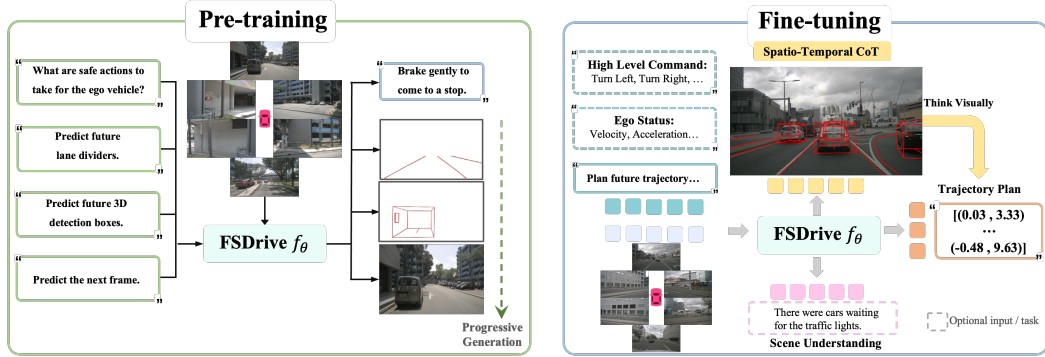

Figure 2: Overview of FSDrive. Taking the currently surround images and task instructions as input, MLLM is trained in the form of next token prediction. MLLM predicts the future spatio-temporal CoT, and then generates trajectory based on the current observation and predicted future.

Our method is directly built upon any existing MLLM that employs ViT-based encoders to convert images into continuous features. We preserve the original MLLM architecture without altering any structural components to maintain compatibility with pretrained weights. The sole modification involves expanding the MLLM's vocabulary by incorporating image tokens of the VQ-VAE into the text codebook, thereby extending the vocabulary's scope from language space to a multimodal space encompassing both visual and textual modalities. This enhancement enables the MLLM to predict image tokens, which can then be converted to image pixels through an VQ-VAE's detokenizer.

**Pre-training for visual understanding.** To effectively preserve the semantic understanding capabilities of the native MLLM during the pre-training stage, as shown in the left part of Figure 2, we follow previous methods [64, 27] by using a VQA task, which is crucial for autonomous vehicles to analyze complex driving scenarios. Given an image-text question-answer pair $(I, L)$, where $I$ represents the surround-view images of the current scene and $L$ denotes the instructional question, model $\mathcal{M}$ generates a corresponding answer $A$:

$$A = \mathcal{M}(I, L). \tag{3}$$

**Pre-training for visual generation.** Inspired by the world models in autonomous driving [30, 77] that generate future frames to learn physical laws, after activating the visual generation capability, we also enable the VLA to predict future frames, thereby capturing the dynamic evolution of the world. Specifically, given an image-instruction pair $(I, L)$, the model predicts the next visual token of the future front-view frame through autoregressive generation:

$$P(q_1, q_2, \ldots, q_{h \cdot w}) = \Pi_{t=1}^{h \cdot w} P_\theta(q_i \mid q_{<i}). \tag{4}$$

The predicted visual tokens are then converted back into image pixels by VQ-VAE's detokenizer. Since future frames naturally exist in video datasets without requiring any labeled data, this approach unlocks the potential to harness abundant video data for improving generation quality.

**Progressive image generation.** However, directly generating complete detailed future scenes may fail to adhere to physical laws [78]. Therefore, during pre-training stage, we propose a progressive, easy-to-hard generation method, incorporating annotated data containing lane divider and 3D detection. Before generating visual tokens of future frames $Q_f$, we leverage the world knowledge of VLA to first reason about visual tokens of lane dividers $Q_l$, which serve as the skeleton of the road scene and define drivable areas to enforce static physical constraints. Subsequently, we reason about visual tokens of 3D bounding boxes $Q_d$, representing motion patterns of key objects to impose dynamic physical constraints. This progressive method sequence explicitly guides the model to infer structural layouts and geometric details of future scenes while enforcing physical laws. By leveraging these intermediate visual reasoning steps as context, the model learns to think visually about the dynamic evolution of scenes, ultimately enabling accurate future prediction:

$$P(Q_f \mid Q_l, Q_d) = \Pi_{t=1}^{h \cdot w} P_\theta(q_i \mid q_{<i}, Q_l, Q_d). \tag{5}$$

## 3.3 Think visually with spatio-temporal CoT

Autonomous driving planning requires not only understanding the current scene but also envisioning potential future developments to achieve forward-looking comprehension. This thinking process should resemble physical world simulation and imagination rather than purely text symbolic logical deduction. Since our model has already learned physical constraints through the progressive generation during pre-training, and considering efficiency, we no longer separately generate lane dividers, 3D detection, and future frames, but instead integrate all these results into a single unified frame. As shown in the right part of Figure 2, here, VLA serves as a world model to generate a unified image frame predicting the future world state: Inspired by visual prompting engineering [53] that draws red circles on images to guide model attention and by VLIPP [78] first predicts future bounding boxes to introduce physical priors when generating future frames, we represent future world spatial relationships through future red lane dividers and 3D detection boxes on the predicted unified frames. These coarse-grained visual cues direct the model's attention toward drivable areas and critical objects in future scenes while enforcing physically plausible constraints. Meanwhile, the temporal relationships are represented by the ordinary future frame, where the dynamic evolution of visual content intuitively characterizes temporal progression and the inherent laws of scene development. Subsequently, spatio-temporal CoT $Q_{CoT}$ serves as an intermediate reasoning step, allowing the VLA to function as an inverse dynamics model that plans trajectory based on current observations and future predictions:

$$P(W_t \mid I_t, Q_{CoT}, opt(T_{com}, T_{ego})) = \Pi_{i=1}^{n} P_\theta(w_i \mid w_{<i}, I_t, Q_{CoT}, opt(T_{com}, T_{ego})). \tag{6}$$

## 3.4 Training strategy

Our FSDrive can be initialized from any existing MLLM (e.g., Qwen2-VL, LLaVA), avoiding training from scratch and saving significant costs. During training, we fully fine-tune the LLM parameters while freezing all encoders. The training process is divided into two stages:

**Stage 1: Unified pre-training.** Our objective is to preserve understanding capabilities of MLLMs through VQA tasks and activate their visual generation capabilities to predict future frames. VQA task data originates from OmniDrive-nuScenes [64]. We incorporate a large volume of unlabeled image data from nuScenes [1] for future frame prediction. To implement progressive easy-to-hard CoT, we integrate nuScenes annotated data to teach the model predicting image-formatted future lane dividers and 3D detection. Finally, we add future frame prediction with CoT datas containing intermediate reasoning steps. All the above understanding and generation tasks are trained together.

**Stage 2: Supervised fine-tuning.** We focus on autonomous driving scene understanding and trajectory planning. Following OmniDrive [64], scene understanding utilizes DriveLM's GVQA [54] dataset. For trajectory planning, we follow VAD [29, 21] using nuScenes, where our spatio-temporal CoT integrates the holistic future scene, explicit lane dividers, and 3D detection results into a single future frame as intermediate reasoning steps. We train these tasks simultaneously using a single model, enabling task-specific predictions during inference through different task prompts.

## 4 Experiments

### 4.1 Experimental settings

**Datasets.** Following the previous methods [29, 13, 4], we evaluate trajectory planning and future frames generation on the nuScenes [1]. The nuScenes contains 1,000 scenes of approximately 20 seconds each captured by a 32-beam LiDAR and six cameras providing 360-degree field of view. Specifically, The dataset provides 28,130 (train), 6,019 (val), and 193,082 (unannotated) samples. Additionally, we conducted experiments on NAVSIM [10], a realistic scenario dataset designed for real-world planning. This dataset aims to highlight challenging driving scenarios involving dynamic changes in driving intent, while deliberately excluding simple situations such as static scenes or constant-speed driving.

Table 1: End-to-end trajectory planning experiments on nuScenes [1]. We evaluated the L2 and collision metrics based on the distinct computational methodologies of ST-P3 [19] and UniAD [21], respectively. * indicates that the ego status is additionally used. VAD [29] and UniAD [21] results are derived from BEV-Planner [34], while the remaining results are sourced from their respective papers.

| Method | ST-P3 metrics | | | | | | | | UniAD metrics | | | | | | | | LLM |
|---|---|---|---|---|---|---|---|---|---|---|---|---|---|---|---|---|---|
| | L2 (m) ↓ | | | | Collision (%) ↓ | | | | L2 (m) ↓ | | | | Collision (%) ↓ | | | | |
| | 1s | 2s | 3s | Avg. | 1s | 2s | 3s | Avg. | 1s | 2s | 3s | Avg. | 1s | 2s | 3s | Avg. | |
| **Non-Autoregressive methods** | | | | | | | | | | | | | | | | | |
| ST-P3* [ECCV22] [19] | 1.33 | 2.11 | 2.90 | 2.11 | 0.23 | 0.62 | 1.27 | 0.71 | - | - | - | - | - | - | - | - | - |
| VAD [ICCV23] [29] | 0.69 | 1.22 | 1.83 | 1.25 | 0.06 | 0.68 | 2.52 | 1.09 | - | - | - | - | - | - | - | - | - |
| VAD* [ICCV23] [29] | 0.17 | 0.34 | 0.60 | 0.37 | 0.04 | 0.27 | 0.67 | 0.33 | - | - | - | - | - | - | - | - | - |
| UniAD [CVPR23] [21] | - | - | - | - | - | - | - | - | 0.59 | 1.01 | 1.48 | 1.03 | 0.16 | 0.51 | 1.64 | 0.77 | - |
| UniAD* [CVPR23] [21] | - | - | - | - | - | - | - | - | 0.20 | 0.42 | **0.75** | 0.46 | 0.02 | 0.25 | 0.84 | 0.37 | - |
| BEV-Planner [CVPR24] [34] | 0.30 | 0.52 | 0.83 | 0.55 | 0.10 | 0.37 | 1.30 | 0.59 | - | - | - | - | - | - | - | - | - |
| BEV-Planner* [CVPR24] [34] | 0.16 | 0.32 | 0.57 | 0.35 | **0.00** | 0.29 | 0.73 | 0.34 | - | - | - | - | - | - | - | - | - |
| PreWorld [ICLR25] [32] | - | - | - | - | - | - | - | - | 0.49 | 1.22 | 2.32 | 1.34 | 0.19 | 0.57 | 2.65 | 1.14 | - |
| **Autoregressive methods** | | | | | | | | | | | | | | | | | |
| ELM [ECCV24] [98] | - | - | - | - | - | - | - | - | 0.34 | 1.23 | 2.57 | 1.38 | 0.12 | 0.50 | 2.36 | 0.99 | BLIP2-2.7B |
| FeD* [CVPR24] [86] | - | - | - | - | - | - | - | - | 0.27 | 0.53 | 0.94 | 0.58 | **0.00** | **0.04** | 0.52 | 0.19 | LLaVA-7B |
| OccWorld [ECCV24] [94] | 0.39 | 0.73 | 1.18 | 0.77 | 0.11 | 0.19 | 0.67 | 0.32 | 0.52 | 1.27 | 2.41 | 1.40 | 0.12 | 0.40 | 2.08 | 0.87 | GPT3-like |
| Doe-1 [arxiv24] [95] | 0.37 | 0.67 | 1.07 | 0.70 | 0.02 | 0.14 | 0.47 | 0.21 | 0.50 | 1.18 | 2.11 | 1.26 | 0.04 | 0.37 | 1.19 | 0.53 | Lumina-mGPT-7B |
| RDA-Driver* [ECCV24] [26] | 0.17 | 0.37 | 0.69 | 0.40 | 0.01 | **0.05** | 0.26 | **0.10** | 0.23 | 0.73 | 1.54 | 0.80 | **0.00** | 0.13 | 0.83 | 0.32 | LLaVA-7B |
| EMMA* [arxiv24] [27] | **0.14** | 0.29 | 0.54 | 0.32 | - | - | - | - | - | - | - | - | - | - | - | - | Gemini 1-1.8B |
| OmniDrive [CVPR25] [64] | 0.40 | 0.80 | 1.32 | 0.84 | 0.04 | 0.46 | 2.32 | 0.94 | - | - | - | - | - | - | - | - | LLaVA-7B |
| OmniDrive* [CVPR25] [64] | **0.14** | 0.29 | 0.55 | 0.33 | **0.00** | 0.13 | 0.78 | 0.30 | - | - | - | - | - | - | - | - | LLaVA-7B |
| **FSDrive (ours)** | 0.28 | 0.52 | 0.80 | 0.53 | 0.06 | 0.13 | 0.32 | 0.17 | 0.40 | 0.89 | 1.60 | 0.96 | 0.07 | 0.12 | 1.02 | 0.40 | Qwen2-VL-2B |
| **FSDrive* (ours)** | **0.14** | **0.25** | **0.46** | **0.28** | 0.03 | 0.06 | **0.21** | **0.10** | **0.18** | **0.39** | 0.77 | **0.45** | **0.00** | 0.06 | **0.42** | **0.16** | Qwen2-VL-2B |
| **FSDrive (ours)** | 0.29 | 0.57 | 0.94 | 0.60 | 0.04 | 0.14 | 0.38 | 0.19 | 0.36 | 1.01 | 1.90 | 1.09 | 0.08 | 0.34 | 1.11 | 0.51 | LLaVA-7B |
| **FSDrive* (ours)** | 0.13 | 0.28 | 0.52 | 0.31 | 0.03 | 0.07 | 0.24 | 0.12 | 0.22 | 0.51 | 0.94 | 0.56 | 0.02 | 0.07 | 0.53 | 0.21 | LLaVA-7B |

Table 2: Performance comparison on NAVSIM navtest using closed-loop metrics. All the methods only use images as input and do not use lidar.

| Method | NC ↑ | DAC ↑ | TTC ↑ | Comf. ↑ | EP ↑ | PDMS ↑ |
|---|---|---|---|---|---|---|
| VADv2 [arXiv24] [3] | 97.2 | 89.1 | 91.6 | **100** | 76.0 | 80.9 |
| UniAD [CVPR23] [21] | 97.8 | 91.9 | 92.9 | **100** | 78.8 | 83.4 |
| DiffusionDrive-Cam [CVPR25] [36] | 97.8 | 92.2 | 92.6 | 99.9 | 78.9 | 83.6 |
| LTF [TPAMI23] [6] | 97.4 | 92.8 | 92.4 | **100** | 79.0 | 83.8 |
| PARA-Drive [CVPR24] [69] | 97.9 | 92.4 | 93.0 | 99.8 | 79.3 | 84.0 |
| LAW [ICLR25] [33] | 96.4 | **95.4** | 88.7 | 99.9 | **81.7** | 84.6 |
| FSDrive (ours) | **98.2** | 93.8 | **93.3** | 99.9 | 80.1 | **85.1** |

Following the previous methods [7, 64], we evaluate scene understanding on DriveLM [54]. This dataset features keyframe descriptions paired with QA annotations covering full-stack autonomous driving (perception, prediction, planning), offering comprehensive language support for development.

**Metrics.** We evaluate trajectory planning using L2 displacement error and collision rate following previous methods [21, 29, 19]. Notably, UniAD [21] computes L2 metrics and collision rate at each timestep, whereas ST-P3 [19] and VAD [29] considers the average of all previous time-steps. For a fair comparison, we adopted these two different calculation methods. Following existing methods [65, 77, 71], we report Fréchet Inception Distance (FID) [17] to measure the future frames generation quality. DriveLM GVQA [54] metrics include language metrics like BLEU, ROUGE_L, and CIDEr for text generation, the ChatGPT Score for open-ended Q&A and accuracy for multiple-choice questions. For NAVSIM [10], we adopt the official metrics for evaluation, especially PDMS.

**Implementation details.** We initialize our model with Qwen2-VL-2B [63] and pre-train it for 32 epochs to enable visual generation while preserving semantic understanding. During fine-tuning (12 epochs on 8 NVIDIA RTX A6000), we use $1 \times 10^{-4}$ learning rate and batch size of 16. We expand the visual codebook of MoVQGAN [92] to the vocabulary of the large language model and use its detokenizer to convert the visual tokens predicted by the large language model to the pixel space.

Table 3: Future frames generation results on the nuScenes [1] dataset.

| Method | DriveGAN [CVPR21 [30]] | DriveDreamer [ECCV24 [65]] | Drive-WM [CVPR24 [66]] | GenAD [CVPR24 [77]] | GEM [CVPR25 [16]] | Doe-1 [arxiv24 [95]] | **FSDrive** |
|---|---|---|---|---|---|---|---|
| **Type** | GAN | Diffusion | Diffusion | Diffusion | Diffusion | Autoregressive | Autoregressive |
| **Resolution** | 256×256 | 128×192 | 192×384 | 256×448 | 576×1024 | 384×672 | 128×192 |
| **FID** ↓ | 73.4 | 52.6 | 15.8 | 15.4 | 10.5 | 15.9 | **10.1** |

Table 4: Results on DriveLM [54] GVQA benchmark.

| Method | Accuracy ↑ | ChatGPT ↓ | BLEU_1 ↑ | ROUGE_L ↑ | CIDEr ↑ | Match ↑ | Final Score ↑ |
|---|---|---|---|---|---|---|---|
| DriveLM baseline [54] | 0.00 | 0.65 | 0.05 | 0.08 | 0.10 | 0.28 | 0.32 |
| Cube-LLM [7] | 0.39 | **0.89** | 0.16 | 0.20 | **0.31** | **0.39** | 0.50 |
| TrackingMeetsLMM [28] | 0.60 | 0.58 | 0.72 | 0.72 | 0.04 | 0.36 | 0.52 |
| SimpleLLM4AD [93] | 0.66 | 0.57 | **0.76** | 0.73 | 0.15 | 0.35 | 0.53 |
| OmniDrive [64] | 0.70 | 0.65 | 0.52 | 0.73 | 0.13 | 0.37 | 0.56 |
| **FSDrive (ours)** | **0.72** | 0.63 | **0.76** | **0.74** | 0.17 | **0.39** | **0.57** |

Table 5: Ablation results of pre-training.

| VQA | Future frames | Future 3D detection | Future lane divider | L2 (m) ↓ | | | | Collision (%) ↓ | | | |
|---|---|---|---|---|---|---|---|---|---|---|---|
| | | | | 1s | 2s | 3s | Avg. | 1s | 2s | 3s | Avg. |
| × | × | × | × | 0.45 | 1.09 | 2.12 | 1.22 | 0.12 | 0.43 | 1.45 | 0.67 |
| ✓ | × | × | × | 0.46 | 1.07 | 2.04 | 1.19 | 0.12 | 0.42 | 1.42 | 0.65 |
| × | ✓ | × | × | **0.39** | 0.96 | 1.71 | 1.02 | 0.10 | 0.38 | **1.32** | 0.60 |
| × | × | ✓ | × | 0.46 | 1.06 | 1.99 | 1.17 | 0.10 | 0.37 | 1.35 | 0.61 |
| × | × | × | ✓ | 0.42 | 0.97 | 1.80 | 1.06 | 0.13 | 0.41 | 1.40 | 0.65 |
| ✓ | ✓ | ✓ | ✓ | **0.39** | **0.91** | **1.63** | **0.98** | **0.09** | **0.36** | 1.33 | **0.58** |

## 4.2 Main results

**End-to-End trajectory planning.** We present trajectory planning performance on nuScenes following previous methods [29, 21] in Table 1. When using ego status, FSDrive surpasses previous SOTA methods using ego status in ST-P3 and UniAD metrics. However, following BEV-Planner [34] findings about ego-status's performance boost, we prioritize non-ego-status evaluations. Compared to non-autoregressive (e.g., UniAD) and autoregressive methods (e.g., OmniDrive), FSDrive demonstrates superior effectiveness. Notably, FSDrive outperforms Doe-1 [95] which also enables vision generation (L2: 0.53 vs. 0.70 and 0.96 vs. 1.26; collision: 0.19 vs. 0.21 and 0.40 vs. 0.53), indicating limitations in their VQ-VAE-based discrete visual features for understanding. For a fair comparison, we also used LLaVA like methods [64, 26, 86, 75]. Under the corresponding settings, FSDrive still has excellent competitiveness, indicating that FSDrive can be widely applied to any existing MLLM.

**Results on NAVSIM.** Table 2 shows the evaluation results for NAVSIM [10]. All approaches rely exclusively on camera input, with no lidar data being used. Achieving a PDMS score of 85.1, FSDrive outperforms prior camera-only methods like LAW [33] and DiffusionDrive-Cam [36], thus showcasing its efficacy in the pseudo closed-loop setting.

**Evaluation of generation results.** Although we generate future frames as CoT for trajectory planning, we still validate visual quality via FID in Table 3. To enable rapid generation for real-time driving, we generate frames at 128×192 resolution. Our autoregressive FSDrive achieves competitive performance against specialized diffusion models. Compared to Doe-1 [95] which employs the vision generation MLLM Lumina-mGPT 7B [37], FSDrive 2B maintains superior advantages, indicating that the visual generation capabilities of MLLM can be effectively unlocked even with minimal data.

**Results on DriveLM dataset.** FSDrive's scene understanding was evaluated on DriveLM in Table 4, achieving 0.57 and outperforming recent methods like Cube-LLM [7] and OmniDrive [64]. This highlights the effectiveness of FSDrive pre-training paradigm for generation and understanding.

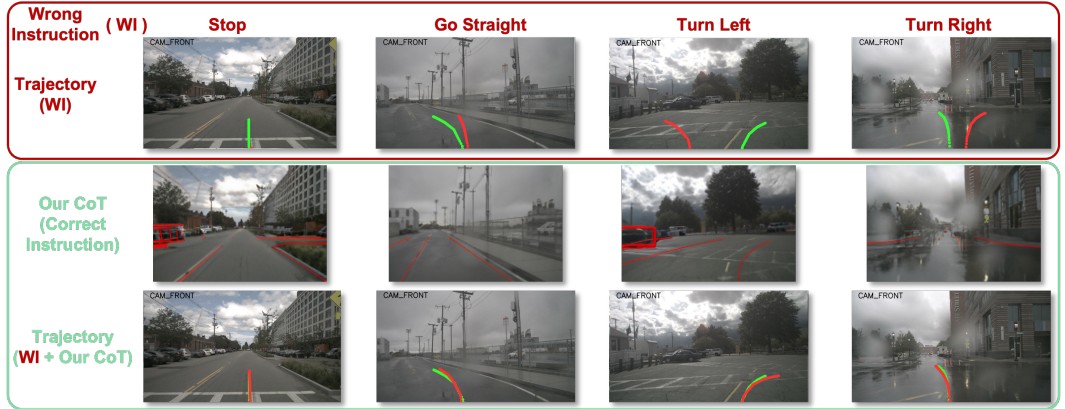

Figure 3: Qualitative analysis of our CoT. The red trajectory is the prediction and the green is the GT.

Table 6: Ablation results of different CoT.

| Type | L2 (m) ↓ | | | | Collision (%) ↓ | | | |
|---|---|---|---|---|---|---|---|---|
| | 1s | 2s | 3s | Avg. | 1s | 2s | 3s | Avg. |
| None | 0.39 | 0.91 | 1.63 | 0.98 | 0.09 | 0.36 | 1.33 | 0.58 |
| Text CoT | 0.39 | 0.92 | 1.61 | 0.97 | 0.10 | 0.29 | 1.21 | 0.53 |
| Image-text CoT | **0.38** | 0.90 | 1.65 | 0.98 | 0.09 | 0.25 | 1.15 | 0.50 |
| Spatio-temporal CoT | 0.40 | **0.89** | **1.60** | **0.96** | **0.07** | **0.12** | **1.02** | **0.40** |

## 4.3 Ablation study

In this section, unless otherwise specified, we evaluate the computing metrics of UniAD [21] based on the Qwen2-VL-2B model [63] and do not use the ego status.

**Qualitative analysis.** We evaluate our CoT's effectiveness in Figure 3. Without spatial-temporal CoT, erroneous navigation inputs caused significant trajectory deviations and potential collisions. Use correct instruction when reasoning our CoT, while still employing wrong instruction for planning. However, FSDrive mitigated instruction errors through observation-based trajectory planning and future prediction, demonstrating its inverse dynamics modeling capability.

Table 7: Ablation experiments of future frames generation.

(a) Ablations of pre-training data volume.

| Data volume | FID↓ |
|---|---|
| None | 29.4 |
| ∼100k | 16.2 |
| ∼200k | **12.7** |

(b) Ablations of progressive method.

| Progressive Method | FID↓ |
|---|---|
| ✕ | 12.7 |
| ✓ | **10.1** |

**Pre-training ablation study.** The impact of pre-training on trajectory planning is summarized in Table 5. Pure VQA tasks show negligible effects. Future frame generation pre-training improves L2 by 16.4% and collisions by 15.8%, validating world-model-based prediction's effectiveness in capturing physical dynamics. 3D detection and lane divider pre-training yield moderate gains in L2/collision metrics respectively. The combined understanding and generation pre-training achieves better performance, demonstrating our unified paradigm's capacity to enhance scene representation and effectively learn physical laws, thereby strengthening spatial understanding capabilities.

**Results of different CoT.** Ablation studies on CoT variants in Table 6 show marginal L2 changes but notable collision rate improvements. Pure text CoT (8.6% improvement) exhibits limited representation capability due to unimodal perception. Compared to text CoT, image-text CoT (combining future frames with textual perception) shows insignificant gains due to the inconsistent modalities

between CoTs. The proposed spatio-temporal CoT achieves 31% improvement, demonstrating that unified image-based reasoning effectively identifies future collision risks.

**Ablation study on generation results.** We conduct ablation studies on future frames generation in Table 7. Table 7a shows that larger pre-training datasets improve MLLM's visual generation capability. Despite being much smaller (200K vs. 100M in previous work [70]), our data achieves more robust visual generation. Scaling datasets may further enhance performance. Table 7b confirms our progressive method improves autoregressive image generation.

## 5 Conclusion

This paper proposes FSDrive, an autonomous driving framework based on spatio-temporal CoT that enables VLAs to think visually. By unifying future scene generation and perception results through intermediate image-form reasoning steps, our FSDrive eliminates the semantic gap caused by cross-modal conversions and establishes an end-to-end visual reasoning pipeline. The VLA serves dual roles: as a world model that predicts future image frames with lane divider and 3D detection, and as an inverse dynamics model that plans trajectory based on both current observations and future predictions. To enable visual generation in VLAs, we present a pretraining paradigm that unifies visual generation and understanding, along with a progressive easy-to-hard visual CoT to enhance autoregressive image generation. Extensive experimental results demonstrate the effectiveness of the proposed FSDrive method, advancing autonomous driving towards visual reasoning.

**Limitations and broader impacts.** Though autonomous driving requires surrounding environmental awareness, considering real-time efficiency, we currently only generate future frames for the front-view. Future work can attempt to generate Surround views to achieve safer autonomous driving. Moreover, more robust visual quality can be achieved in future work through the use of larger training datasets and a more advanced unified paradigm that integrates generation and understanding. In terms of impact, the ethical challenges posed by LLMs extend to autonomous driving. Advances in technology and regulation will drive development of safer, more efficient systems.

**Acknowledgments.** This work was support by the National Natural Science Foundation of China No. 62572385, the Fundamental Research Funds for the Central Universities No. xxj032023020, and CAAI-CANN Open Fund, developed on OpenI Community.

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
