# OpenReview forum: "FutureSightDrive: Thinking Visually with Spatio-Temporal CoT for Autonomous Driving"
_NeurIPS.cc/2025/Conference — NeurIPS 2025 spotlight_

### Official Review · Reviewer_ETeY · 2025-06-22

**Clarity:** 3
**Significance:** 3
**Originality:** 3
**Rating:** 4
**Confidence:** 4

**Summary:**

This paper presents FSDrive, a novel framework for autonomous driving that enables visual reasoning through spatio-temporal Chain-of-Thought (CoT), bridging the gap between perception and planning in a unified visual framework. The experiment part demonstrates that FSDrive achieves good results on the generation, understanding and planing tasks.

**Questions:**

1) Limited architectural novelty: The proposed architecture for generation and understanding tasks largely mirrors existing designs in general vision-language domains. While the application to driving is appreciated, the model structure itself lacks sufficient novelty.
2) Dataset limitation: The claim of using only 0.3% of training data relies solely on experiments conducted on nuScenes. This raises concerns about generalizability. The authors are encouraged to validate their method on additional datasets to strengthen the robustness of their claims.
3) Real-time applicability: The auto-regressive formulation in Equation 6 for action prediction appears computationally expensive and may not satisfy the real-time constraints required in real-world driving systems. Further clarification or optimization is needed here.
4) Insufficient qualitative evaluation: The paper relies on FID to evaluate generative quality, but comparing FID scores across different output resolutions can be misleading. The authors should provide more visual comparisons to convincingly demonstrate the quality of generated future frames.
5) Format of Tab6: A more compact and structured format would help improve the overall clarity of the paper.

**Ethical Concerns:**

["NO or VERY MINOR ethics concerns only"]

**Final Justification:**

The authors have adequately addressed my concerns. The paper’s strengths:
(1) identifying the limitations of textual CoT in autonomous driving,  (2) proposing visual CoT as a principled solution with images as reasoning steps, and (3) providing comprehensive experiments, demonstrating a solid contribution that meets the acceptance bar.

**Limitations:**

Yes

**Quality:**

3

**Strengths And Weaknesses:**

FSDrive presents an improved framework for driving policy prediction, guided by a novel Spatial-Temporal Chain-of-Thought (CoT) methodology. The authors conduct comprehensive experiments to validate the effectiveness of the proposed approach, covering generation, understanding, and policy prediction tasks. I recommend acceptance of this paper, primarily because extending the CoT paradigm into the physical world of autonomous driving is both novel and conceptually sound. The thorough experimental evaluation further supports its potential.

However, there are several weaknesses that should be addressed:

1) Limited architectural novelty: The proposed architecture for generation and understanding tasks largely mirrors existing designs in general vision-language domains. While the application to driving is appreciated, the model structure itself lacks sufficient novelty.
2) Dataset limitation: The claim of using only 0.3% of training data relies solely on experiments conducted on nuScenes. This raises concerns about generalizability. The authors are encouraged to validate their method on additional datasets to strengthen the robustness of their claims.
3) Real-time applicability: The auto-regressive formulation in Equation 6 for action prediction appears computationally expensive and may not satisfy the real-time constraints required in real-world driving systems. Further clarification or optimization is needed here.
4) Insufficient qualitative evaluation: The paper relies on FID to evaluate generative quality, but comparing FID scores across different output resolutions can be misleading. The authors should provide more visual comparisons to convincingly demonstrate the quality of generated future frames.

---

> ### Author Rebuttal · Authors · 2025-07-31
>
> We would like to express our sincere appreciation for your perceptive comments and the rigorous review of our manuscript. Your guidance has been instrumental in refining the scientific rigor and clarity of our paper. We have revised the manuscript accordingly and offer our itemized responses to your queries below:
>
> ---
>
> > **Q1: The model structure itself lacks sufficient novelty.**
>
> **A1:** We sincerely apologize for any confusion caused. We would like to clarify that the core innovation of our work is not to propose a new underlying model architecture. As stated by reviewers 4rpa, TPKG and NHPc, our primary contribution is twofold: firstly, we introduce a novel visual reasoning paradigm specifically for autonomous driving; secondly, we propose a unified and efficient pre-training method to activate the visual generation capabilities of existing LLMs. Furthermore, our proposed unified architecture for generation and understanding can be applied to any existing multi-modal large model, demonstrating its broad applicability and scalability.
>
>
> ---
>
> > **Q2: The claim of using only 0.3% of training data relies solely on experiments conducted on nuScenes. And validating the results on other datasets.**
>
> **A2:**
> | Method           | NC↑  | DAC↑  | TTC↑  | Comf.↑ | EP↑   | PDMS↑ |
> |------------------|------|------|------|-------|------|------|
> | LAW [ICLR 25]    | 96.4 | 95.4 | 88.7 | 99.9  | 81.7 | 84.6 |
> | DiffusionDrive-Cam [CVPR 25] | 97.8 | 92.2 | 92.6 | 99.9  | 78.9 | 83.6 |
> | FSDrive          | 98.2 | 93.8 | 93.3 | 99.9  | 80.1 | 85.1 |
>
> We thank the reviewer for their valuable feedback and sincerely apologize for the lack of clarity that led to this confusion.
> * We would like to clarify a key misunderstanding: our reference to "using only about 0.3% of the data" does not pertain to 0.3% of the nuScenes training set. Instead, this figure is relative to the billion-scale data required by large models that aim to unify visual understanding and generation (e.g., Chameleon, VILA-U). Compared to these models, our approach significantly reduces training costs.
> * Furthermore, we have now evaluated the planning performance of FSDrive on the NAVSIM dataset. FSDrive outperforms previous methods, achieving a PDMS of 85.1, demonstrating excellent generalization capabilities.
>
>
>
> ---
>
> > **Q3: The auto-regressive formulation in Equation 6 may not satisfy the real-time constraints required in real-world driving systems.**
>
> **A3:**
> | Method       | DriveVLM | Senna  | FSDrive |
> |--------------|----------|--------|---------|
> | Decode latency (s)↓ | 0.50    | 0.58   | 0.31    |
>
> FSDrive is efficient for real-time tasks. By integrating vLLM and quantization, it achieves a 0.31s decoding latency, surpassing models like DriveVLM and Senna. This performance establishes a solid foundation for real-time applications, poised to improve further with ongoing developments in large model acceleration.
>
>
> ---
>
> > **Q4: Comparing FID scores across different output resolutions can be misleading. The authors should provide more visual comparisons.**
>
> **A4:** We sincerely thank you for your insightful suggestions. We would like to address your concerns from the following perspectives:
> * Our choice of a lower resolution was a deliberate and carefully considered trade-off between inference efficiency and informational sufficiency. Adopting a higher resolution would significantly compromise the model's inference speed.
> * More importantly, we empirically observed that increasing the generation resolution did not lead to a corresponding improvement in the final planning accuracy. We attribute this to the fact that the number of visual tokens would vastly outnumber the planning tokens, causing the negative impacts to outweigh the potential benefits. Consequently, metrics like FID and fine-grained resolution details are not our primary focus. As detailed in Section 3.3 of our paper, our core objective is to ensure the physical plausibility of the generated scenes and to guide the model's attention, ultimately to support the downstream planning task.
> * It is also worth noting that when compared with DriveDreamer at the same resolution, our method's FID score is substantially better (10.1 vs. 52.6), marking an 81% improvement. This demonstrates that our autoregressive approach remains highly competitive against these diffusion-based methods. We will provide more extensive visual comparisons for a more thorough qualitative analysis in the revised manuscript.
> * Furthermore, as advised, we have included richer visual comparisons in the revised manuscript to better support the qualitative evaluation of our method.
>
> ---
>
> > **Q5: Format of Tab6: A more compact and structured format would help improve the overall clarity of the paper.**
>
> **A5:** Thank you for your valuable suggestions. We have reformatted Table 6 for better conciseness.

---

> > ### Comment · Reviewer_ETeY · 2025-08-02
> >
> > Thank you for your response. I have reviewed it and am satisfied that my concerns have been adequately addressed. I appreciate the additional clarifications and hope the rebuttal discussions are reflected in the final version.

---

> > > ### Author Response · Authors · 2025-08-02
> > >
> > > Thank you for your thoughtful review and for confirming that our concerns have been addressed—I sincerely appreciate your time and constructive feedback. As requested, we will carefully integrate the discussions from this rebuttal into the final manuscript to ensure clarity and completeness.

---

### Official Review · Reviewer_NHPc · 2025-06-23

**Clarity:** 3
**Significance:** 3
**Originality:** 3
**Rating:** 5
**Confidence:** 4

**Summary:**

This paper proposes FSDrive, an autonomous driving framework based on Spatio-Temporal Chain-of-Thought (CoT), which addresses the issues of ambiguous spatio-temporal relationships and fine-grained information loss caused by discrete textual CoT in existing Vision-Language Models (VLMs). The key contributions include: Visual Reasoning: By generating future frames (with lane dividers and 3D detection boxes) as intermediate reasoning steps, the model performs visual simulation akin to human cognition rather than symbolic logic. Unified Pretraining: Joint optimization of visual generation (future frame prediction) and understanding (VQA), coupled with progressive generation (coarse physical constraints first, then fine details). Experimental Validation: On nuScenes and DriveLM datasets, FSDrive outperforms SOTA methods (e.g., UniAD, Doe-1) in trajectory planning, future frame generation, and scene understanding.

**Questions:**

How does the model handle noisy annotations (e.g., occlusions) for lane dividers and 3D boxes?

**Ethical Concerns:**

["NO or VERY MINOR ethics concerns only"]

**Final Justification:**

I have reviewed the author response. The rebuttal addressed the main concerns I raised, and the clarifications provided were reasonable. I have no further concerns and will maintain my original positive score.

**Limitations:**

yes

**Quality:**

3

**Strengths And Weaknesses:**

Strengths
1. Clearly identifies limitations of textual CoT in autonomous driving (loss of physical laws due to over-abstraction) and proposes visual reasoning as a principled solution.
2. The core innovation lies in elevating text CoT to visual CoT, directly using images as intermediate reasoning steps, which better suits the spatio-temporal continuity requirements of autonomous driving.
3. Experimental validation is comprehensive and robust.

Weaknesses
1. Inference speed for generating future frames is unreported. Compliance with real-time requirements (e.g., >10Hz) remains unclear.
2. Only front-view future frames are generated, while autonomous driving requires 360° environmental awareness. This may limit real-world deployment safety.

---

> ### Author Rebuttal · Authors · 2025-07-31
>
> We are deeply grateful for the time and effort you have dedicated to reviewing our manuscript. Your constructive suggestions are immensely valuable and have provided us with an excellent opportunity to substantially enhance our work. We have carefully considered each of your points and our detailed responses are provided below:
>
> ---
>
> > **Q1: Inference speed for generating future frames is unreported.**
>
> **A1:**
> | Method       | DriveVLM | Senna  | FSDrive |
> |--------------|----------|--------|---------|
> | Decode latency (s)↓ | 0.50    | 0.58   | 0.31    |
>
> FSDrive is efficient for real-time tasks. By integrating vLLM and quantization, it achieves a 0.31s decoding latency, surpassing models like DriveVLM and Senna. This performance establishes a solid foundation for real-time applications, poised to improve further with ongoing developments in large model acceleration.
>
>
> ---
>
> > **Q2: Only front-view future frames are generated, while autonomous driving requires 360° environmental awareness.**
>
> **A2:** We sincerely thank the reviewer for their insightful comments and constructive feedback. We offer the following clarifications:
> * In line with established previous methods [1,2], our approach generates only the front view, as this view contains the essential information.  This design choice allows us to achieve a practical trade-off between inference efficiency and informational sufficiency.
> * We would like to clarify that while the proposed method generates future frames exclusively for the front view, its input consists of images from six cameras, encompassing a full 360° perception of the environment, allowing  model to leverage information from all other views. Furthermore, we have proactively and explicitly acknowledged this point in the "Limitations and broader impacts" section on page 9 of our paper and have designated its extension as a direction for future work.
>
>
>
> ---
>
> > **Q3: Noisy annotations (e.g., occlusions) for lane dividers and 3D boxes.**
>
>
> **A3:** We directly use the official annotations from the nuScenes dataset. The dataset creators have already addressed ambiguous cases during the collection process and provide reliable ground truth labels even under conditions of occlusion. And when the object is occluded in the front view, since our input is in the surround-view, the model can still obtain the information of the occluded object from other perspectives.
>
> ## References
> [1] Hassan, Mariam et al. “GEM: A Generalizable Ego-Vision Multimodal World Model for Fine-Grained Ego-Motion, Object Dynamics, and Scene Composition Control.” CVPR 2025.
>
> [2] Yang, Jiazhi et al. “Generalized Predictive Model for Autonomous Driving.” CVPR 2024.

---

> > ### Comment · Reviewer_NHPc · 2025-08-01
> >
> > Thank you for the response. I have read your reply carefully and found that it has addressed the concerns I previously raised. I appreciate the additional analysis and clarifications. I hope that the discussions during the rebuttal can be reflected in the final version, including the inclusion of the inference speed experiments and other supplementary points.

---

> > > ### Author Response · Authors · 2025-08-02
> > >
> > > Thank you for your constructive suggestions. We will incorporate the content discussed in the rebuttal into the final version.

---

### Official Review · Reviewer_TPKG · 2025-06-29

**Clarity:** 3
**Significance:** 2
**Originality:** 3
**Rating:** 4
**Confidence:** 3

**Summary:**

This paper addresses the limitations of current visual language models (VLMs) in autonomous driving, particularly the reliance on discrete, text-based Chain-of-Thought (CoT) reasoning. The authors argue that such symbolic, language-based intermediate steps can lead to ambiguity in representing spatio-temporal relationships and result in the loss of fine-grained visual information, which is critical for safe and effective autonomous driving.

To overcome these issues, the paper introduces a spatio-temporal CoT reasoning method that enables VLMs to "think visually" rather than abstracting visual information into text. In this approach, the VLM acts as a world model, generating unified image frames that predict future world states. These frames include both perception results (like lane dividers and 3D object detections) to capture spatial relationships, and ordinary future frames to represent temporal evolution. This visual reasoning process serves as an intermediate step, allowing the VLM to function as an inverse dynamics model for trajectory planning.

**Questions:**

- The authors should explain why the improvement is only marginal under fair comparison, as this is critical for assessing the significance of the proposed methods.
- A fair comparison should be provided in Table 2 to validate the effectiveness of the proposed methods.

**Ethical Concerns:**

["NO or VERY MINOR ethics concerns only"]

**Final Justification:**

After considering the rebuttal and the comments from other reviewers, I find that most of my concerns have been addressed. Accordingly, I am updating my score to the borderline accept.

**Limitations:**

yes

**Paper Formatting Concerns:**

No.

**Quality:**

2

**Strengths And Weaknesses:**

Strengths:
1. The experimental evaluation is comprehensive, covering multiple relevant tasks (trajectory planning, future frame generation, scene understanding).
2. The proposed spatio-temporal CoT method is clearly described.

Weaknesses:
1. Although the overall training process is intuitive and complete, the improvement in trajectory planning based on LLaVA-7B over other state-of-the-art methods is only marginal. Given the sophistication of the proposed methods, the performance gains do not seem commensurate. Furthermore, while the CoT process has been enhanced, the improvement over the DriveLM benchmark remains limited.
2. As shown in Table 2, FID should be compared at the same resolution. Training with a lower resolution may yield a lower FID score, but this does not necessarily indicate better performance.

---

> ### Author Rebuttal · Authors · 2025-07-31
>
> Thank you for your thoughtful review and constructive feedback. Your comments were invaluable in helping us strengthen the manuscript. Please find our point-by-point responses below:
>
> ---
>
> > **Q1: The improvement in trajectory planning based on LLaVA-7B over other SOTA methods is only marginal. And the improvement achieved on the DriveLM benchmark is still relatively limited.**
>
> **A1:**
> | Method           | L2 (1s) | L2 (2s) | L2 (3s) | L2 Avg | Collision (1s) | Collision (2s) | Collision (3s) | Collision Avg |
> |------------------|---------|---------|---------|--------|----------------|----------------|----------------|---------------|
> | OmniDrive[CVPR25] (LLaVA-7B) | 0.59    | 1.34    | 2.33    | 1.42   | 0.28           | 1.18           | 2.74           | 1.40          |
> | OmniDrive*[CVPR25] (LLaVA-7B)       | 0.19    | 0.56    | 1.19    | 0.65   | 0.10           | 0.40           | 1.43           | 0.64          |
> | FSDrive(LLaVA-7B)          | 0.36    | 1.01    | 1.90    | 1.09   | 0.08           | 0.34           | 1.11           | 0.51          |
> | FSDrive*(LLaVA-7B)         | 0.22    | 0.51    | 0.94    | 0.56   | 0.02           | 0.07           | 0.53           | 0.21          |
> We apologize for the confusion and would like to offer the following clarification:
> * As shown in Table 1 of the paper under the ST-P3 metric, our method without ego status achieves a 29% reduction in L2 error (0.60 vs. 0.84) and an 80% reduction in collision rate (0.19 vs. 0.94) compared to OmniDrive (LLaVA-7B), which are very significant improvements. While it's true that all methods perform well when ego status is used, the ST-P3 metric is not stringent enough to reveal meaningful differences.
> * Therefore, for a clearer comparison, we evaluated OmniDrive using UniAD's metrics in the table above. Without ego status, our method achieves a 23% lower L2 error (1.09 vs. 1.42) and a 64% lower collision rate (0.51 vs. 1.40). With ego status, we still see a 14% L2 reduction (0.56 vs. 0.65) and a 67% collision rate reduction (0.21 vs. 0.64). This evidence confirms that our improvements are not "marginal," but rather robust and substantial across different evaluation standards.
> * It is important to note that on the DriveLM benchmark, we compare our 2B model against the 7B models from competing methods. In VQA tasks like this, which typically benefit significantly from model size, our smaller model achieves superior performance.
>
>
>
>
>
> ---
>
> > **Q2: Training with a lower resolution may yield a lower FID score, but this does not necessarily indicate better performance.**
>
>
>
> **A2:** We sincerely thank you for your insightful suggestions. We would like to address your concerns from the following perspectives:
> * Our choice of a lower resolution was a deliberate and carefully considered trade-off between inference efficiency and informational sufficiency. Adopting a higher resolution would significantly compromise the model's inference speed.
> * More importantly, we observed that increasing the generation resolution did not lead to a corresponding improvement in the final planning accuracy. Consequently, metrics like FID and fine-grained resolution details are not our primary focus. As detailed in Section 3.3 of our paper, our core objective is to ensure the physical plausibility of the generated scenes and to guide the model's attention, ultimately to support the downstream planning task.
> * It is also worth noting that when compared with DriveDreamer at the same resolution, our method's FID score is substantially better (10.1 vs. 52.6), marking an 81% improvement. This demonstrates that our autoregressive approach remains highly competitive against these diffusion-based methods. We will provide more extensive visual comparisons for a more thorough qualitative analysis in the revised manuscript.

---

> > ### Comment · Reviewer_TPKG · 2025-08-02
> >
> > Thanks for your rebuttal. After reviewing the rebuttal along with the other reviewers' comments, I believe it addresses most of my concerns. I will raise my score to the borderline accept.

---

> > > ### Author Response · Authors · 2025-08-02
> > >
> > > We are delighted to hear that our rebuttal has resolved the concerns raised. We are sincerely grateful for the reviewer's acknowledgment and insightful comments.
> > > We would like to assure you that we will diligently incorporate all the revisions and discussion points we committed to in the rebuttal into the final manuscript.
> > > Once again, we thank you for your invaluable support and guidance.

---

### Official Review · Reviewer_4rpa · 2025-07-03

**Clarity:** 3
**Significance:** 2
**Originality:** 2
**Rating:** 5
**Confidence:** 3

**Summary:**

This paper proposes a spatio-temporal Chain-of-Thought (CoT) method to predict future trajectories in autonomous driving. In this framework, the Visual Language Model (VLM) functions as an inverse dynamics model, planning trajectories based on current observations and its own predicted future states. The training process is divided into pre-training and fine-tuning stages and is conducted without altering the existing MLLM architecture. During the pre-training phase, a "progressive generation" technique is employed to ensure the future predictions adhere to physical laws. The authors report that this method achieves state-of-the-art (SOTA) performance compared to existing approaches.

**Questions:**

1. I am curious about the necessity of the VQA task. Could the authors provide the results of an ablation study using only the three generation-related tasks (Future frames, Future 3D detection, and Future lane divider) without the VQA task?
2. Would it be challenging to produce results on other datasets? For instance, evaluating on Argoverse 2, which is often used alongside nuScenes, would significantly strengthen the claims of robustness.
3. Considering that other SOTA papers like OmniDrive report results specifically for intersections, why was this critical scenario analysis omitted? Would it be possible to conduct and share additional experiments for such cases?

**Ethical Concerns:**

["NO or VERY MINOR ethics concerns only"]

**Final Justification:**

The authors rebuttal has addressed my initial concerns. I am now convinced that most of their claims are well-supported by clear experimental evidence. Accordingly, I am raising my score from borderline accept to accept.

**Limitations:**

1. The method's ability to operate in real-time is a significant concern, making its practical application in live driving scenarios challenging.
2. There is no dedicated module to handle cases where the CoT contains errors, which could compromise safety.
3. The results are limited to a single dataset, raising concerns about how well the model will generalize to diverse, unseen environments.

**Quality:**

3

**Strengths And Weaknesses:**

Strengths
1. The method is cost-effective as it does not modify the existing MLLM architecture, only requiring an expansion of the vocabulary during the pre-training stage.
2. The progressive generation technique effectively incorporates physical laws, ensuring the model predicts more realistic future scenarios.

Weaknesses:
1. There is a significant concern regarding real-time performance. The paper provides no metrics on inference latency, and it seems challenging to apply the prediction results immediately in a live driving context.
2. The paper does not present failure cases or incorrect examples of the generated CoT, which makes it difficult to assess the model's limitations from the qualitative results.
3. The generated CoT is an output of the model's inference, not the fundamental reason for the decision itself. Therefore, it is difficult to consider it a true basis for the model's actions.

---

> ### Author Rebuttal · Authors · 2025-07-31
>
> Thank you for your valuable time and insightful comments. Your feedback has been crucial for improving our manuscript. We have addressed your comments point-by-point as follows:
>
> ---
>
> > **Q1: The paper did not provide the metrics of inference latency and real-time performance.**
>
> **A1:**
> | Method       | DriveVLM | Senna  | FSDrive |
> |--------------|----------|--------|---------|
> | Decode latency (s)↓ | 0.50    | 0.58   | 0.31    |
>
> FSDrive is efficient for real-time tasks. By integrating vLLM and quantization, it achieves a 0.31s decoding latency, surpassing models like DriveVLM and Senna. This performance establishes a solid foundation for real-time applications, poised to improve further with ongoing developments in large model acceleration.
>
>
>
> ---
>
> > **Q2: The paper does not provide failure cases of the generated CoT, and there is no dedicated module to handle cases where the CoT contains errors.**
>
> **A2:**
> 1. On CoT Failure Cases
>
> We thank the reviewer for this insightful suggestion. We will incorporate a failure case analysis in the revision. Our key observations are twofold:
>
> * Our model can achieve correct planning even with minor imperfections in the visual CoT. It extracts the essential gist of the future scene, a finding that corroborates GenHancer [1], which states, "imperfect visual generation can still be beneficial for understanding."
> * Due to the data distribution challenges in complex scenarios, while a degraded visual CoT can lead to trajectory deviations, the model's sometimes success under these conditions indicates it has learned a crucial fusion strategy. It effectively balances the generated future (CoT) with current observations, rather than relying solely on the former.
>
> 2. On Handling Erroneous CoT
>
> While FSDrive does not include a dedicated module for CoT error correction, its design provides robustness. This resilience stems from two key aspects of our approach:
>
> * Dual-Source Fusion: The final planning policy is not solely dependent on the CoT. It learns to fuse and balance two information streams—current observations and the generated future—thereby mitigating the impact of an failed CoT.
> * Progressive Pre-training: Our design proactively reduces CoT errors from the outset. As detailed in Sec. 3.2, the pre-training phase employs a progressive, easy-to-hard settings. This strategy systematically improves the model's predictive accuracy, thus minimizing the generation of erroneous CoTs.
> * Future Work: We agree that an explicit safety module is a valuable future direction. For instance, the system could discard a low-quality CoT (e.g., below an FID threshold) and trigger a re-planning mechanism, adding another layer of safety.
>
>
>
>
>
>
>
> ---
>
> > **Q3: The generated CoT is an output of the model's inference, not the fundamental reason for the decision itself.**
>
> **A3:** As argued in [2], a CoT is an essential mechanism for tackling complex tasks, rather than a mere output. To address your concerns, we elaborate on this from the following three aspects:
> * Sequential Generation Process: As shown in Equation 6, our model operates under an auto-regressive paradigm. The generation of our CoT and the subsequent trajectory planning is a sequential process. Critically, the prediction of trajectory planning tokens is conditioned on the previously generated CoT tokens, mirroring the standard CoT generation in LLMs[3].
> * Quantitative Impact: Our ablation study (Table 5) confirms the CoT's critical role. Its removal significantly degrades performance, increasing the collision rate from 0.40% to 0.58%, which directly proves its impact on decision-making.
> * Qualitative Analysis: As shown in Figure 3, when challenged with an incorrect command, our model leverages the CoT to mitigate the faulty instruction and execute a safe maneuver.  This demonstrates the CoT's function as a safety-critical reasoning layer, not a passive output.
>
>
>
> ---
>
> > **Q4: Provide the results of an ablation study using only the three generation-related tasks without the VQA task.**
>
> **A4:**
> | Method                     | L2 (1s) | L2 (2s) | L2 (3s) | L2 Avg | Collision (1s) | Collision (2s) | Collision (3s) | Collision Avg |
> |----------------------------|---------|---------|---------|--------|-----------------|-----------------|-----------------|---------------|
> | w/o pre-training           | 0.45    | 1.09    | 2.12    | 1.22   | 0.12            | 0.43            | 1.45            | 0.67          |
> | w/ generation pre-training | 0.38    | 0.95    | 1.70    | 1.01   | 0.09            | 0.37            | 1.35            | 0.60          |
> | w/ generation & understanding pre-training | 0.39 | 0.91 | 1.63 | 0.98 | 0.09 | 0.36 | 1.33 | 0.58 |
>
> We will address your concerns from the following two aspects:
> * VQA pre-training was specifically incorporated to boost performance on the DriveLM benchmark, successfully raising the score from 0.55 to 0.57. Notably, our model without VQA pre-training (0.55) still surpasses the OmniDrive (0.53).
> * Our ablation study reveals that VQA's impact on planning-centric tasks is modest.  Its removal leads to only a minor performance decrease of 0.02-0.03, suggesting it provides a slight enhancement but is not a dominant factor.  We maintain that exploring the synergy between understanding and generation remains a vital research direction for unified generation and understanding LLMs.
>
>
>
> ---
>
> > **Q5: The evaluation results on other datasets, such as Argoverse 2.**
>
> **A5:**
> | Method           | NC↑  | DAC↑  | TTC↑  | Comf.↑ | EP↑   | PDMS↑ |
> |------------------|------|------|------|-------|------|------|
> | LAW [ICLR 25]    | 96.4 | 95.4 | 88.7 | 99.9  | 81.7 | 84.6 |
> | DiffusionDrive-Cam [CVPR 25] | 97.8 | 92.2 | 92.6 | 99.9  | 78.9 | 83.6 |
> | FSDrive          | 98.2 | 93.8 | 93.3 | 99.9  | 80.1 | 85.1 |
>
> The Argoverse 2 dataset is not designed for planning tasks; consequently, prior works have generally not evaluated planning performance on it. To bridge this gap, we evaluated FSDrive on the NAVSIM dataset. Our method surpasses existing approaches, achieving a final PDMS score of 85.1, demonstrating excellent generalization capabilities.
>
>
>
>
> ---
>
> > **Q6: Report results specifically for intersections.**
>
> **A6:**
> | Method           | Intersection (1s) | Intersection (2s) | Intersection (3s) | Intersection Avg |
> |------------------|--------------------|--------------------|--------------------|-------------------|
> | OmniDrive [CVPR25] | 0.93              | 3.65              | 8.28              | 4.29             |
> | FSDrive          | 0.85              | 3.24              | 6.75              | 3.61             |
>
> We sincerely thank the reviewer for this insightful suggestion. To clarify, the 'Intersection' metric represents a fine-grained breakdown of the overall 'Collision'. This is the primary reason we did not report it individually in the initial submission. As requested, we now present the results for the Intersection metric. As shown above, our method outperforms OmniDrive by 16% on this metric as well, further demonstrating the robustness of our approach. We will add this result to the revised manuscript.
>
> ## References
> [1] Ma, Shijie et al. “GenHancer: Imperfect Generative Models are Secretly Strong Vision-Centric Enhancers.” ICCV 2025.
>
> [2] Feng, Guhao et al. “Towards Revealing the Mystery behind Chain of Thought: a Theoretical Perspective.” NeurIPS 2023.
>
> [3] Wei, Jason, et al. "Chain-of-thought prompting elicits reasoning in large language models." NeurIPS 2022.

---

> > ### Comment · Reviewer_4rpa · 2025-08-03
> >
> > Thank you for providing a detailed response and the additional experimental results. After reviewing your rebuttal, I find that most of my concerns have been well addressed. The new results provide strong support for the claims made in the paper. Therefore, I will raise my score to accept. I strongly encourage you to incorporate the new findings and discussions from your response into the final version of the manuscript.

---

> > > ### Author Response · Authors · 2025-08-03
> > >
> > > Thank you sincerely for carefully reviewing our rebuttal and providing such positive feedback. We are very grateful that you recognize the supplementary experimental results and related discussions we provided, and that you are willing to increase the score to indicate your acceptance. We fully agree with your suggestions and will incorporate the relevant discussions from the rebuttal into the final version of the paper. Thank you again for your valuable time and constructive comments, which have greatly improved the quality of our paper.

---

### Decision · Program_Chairs · 2025-09-17

**Decision:**

Accept (spotlight)

**Comment:**

This paper introduces FutureSightDrive, which replaces text-based Chain-of-Thought reasoning in autonomous driving with visual reasoning through generated future image frames containing lane markers and 3D detection boxes as intermediate reasoning steps for trajectory planning. Four reviewers initially provided mixed feedback with concerns about real-time performance, front-view-only generation, and limited generalization, but all ultimately supported acceptance after the authors provided a comprehensive rebuttal with additional experiments demonstrating 0.31s decode latency, NAVSIM dataset results, and intersection-specific metrics. The paper's key strengths include a novel application of visual CoT that addresses limitations of abstract text reasoning, practical real-time viability, and cost-effective implementation requiring no architectural changes to existing MLLMs, while the main limitation remains the restriction to front-view generation. Despite modest architectural novelty, the visual reasoning paradigm represents meaningful progress in autonomous driving applications of vision-language models with adequate experimental validation.